# STAR: Spatial-Temporal Tracklet Matching for Multi-Object Tracking

**Xuewei Bai**[1]    **Yongcai Wang**[1]*    **Deying Li**[1†]    **Haodi Ping**[2‡]    **Chunxu Li**[1,3]

[1] School of Information, Renmin University of China
[2] School of Computer Science, Beijing University of Technology
[3] China Waterborne Transport Research Institute
{bai_xuewei,ycw,deyingli}@ruc.edu.cn
haodi.ping@bjut.edu.cn
lichunxu@wti.ac.cn

## Abstract

Existing tracking-by-detection Multi-Object Tracking methods mainly rely on associating objects with tracklets using motion and appearance features. However, variations in viewpoint and occlusions can result in discrepancies between the features of current objects and those of historical tracklets. To tackle these challenges, this paper proposes a novel Spatial-Temporal Tracklet Graph Matching paradigm (STAR). The core idea of STAR is to achieve long-term, reliable object association through the association of Tracklet Clips (TCs). TCs are segments of confidently associated multi-object trajectories, which are linked through graph matching. Specifically, STAR initializes TCs using a Confident Initial Tracklet Generator (CITG) and constructs a TC graph via Tracklet Clip Graph Construction (TCGC). In TCGC, each object in a TC is treated as a vertex, with the appearance and local topology features encoded on the vertex. The vertices and edges of the TC graph are then updated through message propagation to capture higher-order features. Finally, a Tracklet Clip Graph Matching (TCGM) method is proposed to efficiently and accurately associate the TCs through graph matching. STAR is model-agnostic, allowing for seamless integration with existing methods to enhance their performance. Extensive experiments on diverse datasets, including MOTChallenge, DanceTrack, and VisDrone2021-MOT, demonstrate the robustness and versatility of STAR, significantly improving tracking performance under challenging conditions. The code is available at `https://github.com/baixuewei430-dotcom/STAR`.

## 1 Introduction

Multi-object tracking (MOT) is a longstanding task in computer vision [1, 2, 3, 4], generally divided into two paradigms: tracking-by-detection (TBD) and joint detection-and-tracking (JDT). Currently, TBD [5, 6, 7, 8, 9] generally outperforms JDT in terms of accuracy. The core task in TBD involves effectively extracting object features and designing accurate association strategies to assign stable IDs to the same object. However, existing methods still encounter challenges in feature extraction and data association in difficult scenarios, such as crowded environments or frequent occlusions.

---

*Dr. Wang is supported in part by the National Natural Science Foundation of China Grant No.61972404, Public Computing Cloud, Renmin University of China, and the Blockchain Lab. School of Information, Renmin University of China.

†Dr. Li is supported in part by the National Natural Science Foundation of China Grant No.12071478.

‡Dr. Ping is supported by the Beijing Natural Science Foundation under Grant 4244076.

Reliable object association in crowded or occluded scenarios poses challenges for both feature extraction and association. Existing methods utilize a variety of features, including object appearance [10, 11, 12, 13], motion [14, 15, 16, 17, 18], temporal [19, 20, 21], and neighborhood topology features [22, 23, 24, 25]. However, these features may not be sufficiently distinctive in such challenging environments. For association, methods mainly employ bipartite matching [26, 27, 28, 29, 30] and graph matching [31, 32]. While graph matching provides higher accuracy, its computational cost increases rapidly with the number of objects and is highly reliant on the distinctiveness of features.

To address the challenges of object association in crowded and occluded scenarios, we propose a novel **S**patial-**T**emporal tr**A**cklet g**R**aph matching paradigm (STAR). The key aspect of STAR is its ability to maintain reliable long-term object associations, even when objects are occasionally obstructed. The central idea is to enhance feature distinctiveness by utilizing Tracklet Clips instead of individual object instances. A Tracklet Clip (TC) refers to a segment of confidently associated trajectories of multiple objects, which captures the appearance, spatial, and temporal features of an object, making it more distinctive in the feature space. Additionally, we further enhance the features of TCs by integrating topology information and high-order features through graph neural networks.

More specifically, STAR consists of three components. (1) The Confident Initial Tracklet Generator (CITG) uses a dynamically adjusted IoU-based method to produce initial tracklet segments from the input video, ensuring consistency by adaptively separating tracklets of the same object during occlusion. (2) The Tracklet Clip Graph Construction component rearranges tracklet segments by timestamps, divides them into tracklet clips (TCs), and converts them into feature graphs. Each tracklet within a TC is treated as a vertex, while the relationships between these vertices are modeled as edges. Vertex features integrate appearance, topology, and temporal information, with message propagation used to capture higher-order features. (3) The Tracklet Clip Graph Matching (TCGM) module enables fast and accurate TC association through graph matching. The main contributions of this work are summarized as follows:

1. STAR innovatively reformulates the object association problem into the construction and matching of Tracklet Clip feature graphs, thus effectively addressing occlusions.

2. The proposed Tracklet Clip Graph Construction (TCGC) method enhances distinctiveness and robustness of each object's feature by constructing multi-object spatial-temporal feature graphs.

3. Our Tracklet Clip Graph Matching (TCGM) approach employs tracklet-level graph matching to not only enhance matching accuracy but also overcome the efficiency bottlenecks associated with traditional frame-level graph matching.

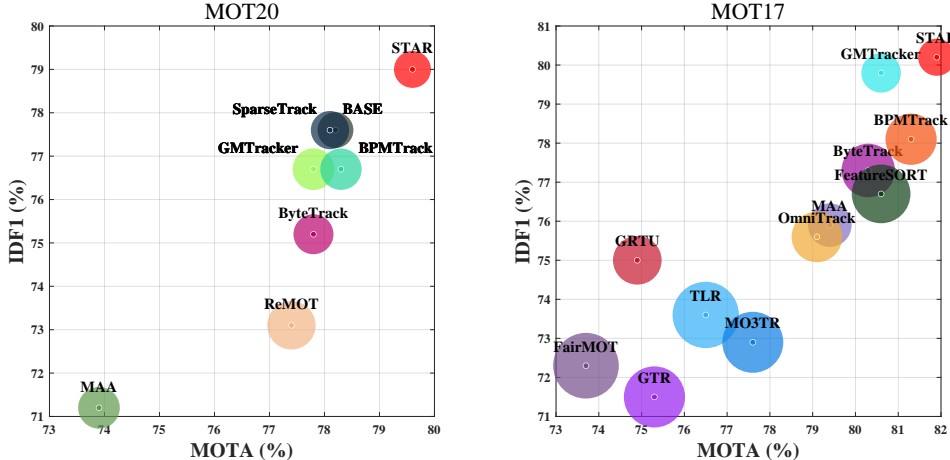

Figure 1: Comparison on MOT17 and MOT20. The x-axis represents MOTA, the y-axis represents IDF1, and the bubble size indicates the number of ID Switches (smaller is better).

# 2 Related Work

MOT has gained significant attention across industry and academia. We review relevant methods focusing on feature extraction and data association approaches.

## 2.1 Feature Extraction

Features fall into four categories: motion, appearance, temporal, and topology. We examine how existing methods address feature discontinuities caused by viewpoint changes.

**Motion Feature.** Basic motion features (position, velocity, bounding box dimensions) become unreliable with irregular camera movement. OCSORT [33] modifies the Kalman filter [34] to prioritize detection results. StrongSORT [35] and MAT [36] use regression to connect fragmented trajectories. While BoT-SORT [11] and MAA [15] compensate for camera motion through coordinate transformations, they remain computationally expensive and struggle in crowded scenes.

**Appearance Feature.** Deep learning models extract high-dimensional appearance vectors [37] that complement motion features. FairMOT [10] enhances feature extraction, while SimpleTrack [38] and BoT-SORT [11] optimize feature combinations. Xie et al. [39] develop region-based networks for fine-grained features. However, these methods perform poorly with UAV footage where object textures are less distinct and undergo perspective deformation.

**Temporal Feature.** These features model frame-to-frame dependencies. CenterTrack [19] and STTA [20] employ temporal attention, while McLaughlin et al. [40] use recurrent networks. Zhang et al. [21] propose orderless representations for better temporal modeling. These approaches, however, are sensitive to occlusions and often neglect multi-object relationships.

**Topology Feature.** Graph-based structures capture object relationships. GSM [22] builds directed graphs based on relative positions. GTAN [23] creates graphs between detections and trajectories but overlooks intra-frame relationships. [25] introduces topology features that remain stable under viewpoint changes but often ignore temporal information.

Existing methods underutilize tracklet-level features, limiting their effectiveness against occlusions and viewpoint changes.

## 2.2 Data Association

Data association connects current object features with previous tracklet features through bipartite or graph matching.

**Bipartite Matching.** This approach treats association as a linear assignment problem. ByteTrack [41] uses thresholding while retaining low-confidence detections. This two-stage association has become standard (used in [26, 27, 17, 28]). Some methods extend this approach: [29] uses three-stage association, while LG-Track [30] employs four stages. However, multi-stage approaches that don't consider all tracklet-detection pairs simultaneously can introduce identity switches.

**Graph Matching.** This approach formulates data association as a graph problem, where vertices represent objects or tracklets and edges capture their relationships. GM [31] was the first to apply this method in MOT, significantly improving association accuracy. GPM [9] pioneers the abstraction of the multi-object tracking problem into frame-level point set matching. SuperGlue [42] combines graph matching with deep learning but mainly focuses on keypoint relationships, neglecting important intra-frame connections. GMTracker [32] enhances convergence speed by replacing the Sinkhorn layer with a graph matching network. However, graph matching methods have notable limitations: computational costs increase rapidly with the number of objects, and accuracy is highly reliant on the distinctiveness of features, which restricts their scalability for large-scale MOT applications.

To address these challenges, we propose STAR, which utilizes spatial-temporal information to create distinctive and robust TC feature graphs, while efficiently reducing computational bottlenecks and maintaining accuracy through effective TC feature graph matching.

# 3 Methodology

## 3.1 Problem Definition and Overview

Given an input video sequence of $L$ frames, the detected objects in frame $t$ are represented as $\mathbb{I}^t = \{o_1, o_2, \ldots, o_{n_t}\}$, where $n_t$ is the number of objects. Each detection $o_i$ is a tuple $o_i = \{pos_i, \mathbf{f}_i, score_i, t_i\}$, with $pos_i$ as the position, $\mathbf{f}_i$ as the appearance feature, $score_i$ as the confidence score, and $t_i$ as the timestamp. The goal of multi-object tracking (MOT) is to generate trajectories for all objects by matching detections that refer to the same object, represented as $T^* = \{T_1, T_2, \ldots, T_n\}$. An overview of STAR is shown in Figure 2. The detection set is processed by the Confident Initial Tracklet Generator (CITG) to produce Confident Initial Tracklets (CITs), denoted as $G = \{g_1, g_2, \ldots, g_k\}$. It is important to ensure that CITs are generated with a strict object association strategy, so each CIT corresponds to the same object. Due to occlusions, a single object's trajectory may be split into multiple CITs, leading to a total number $k$ of CITs that is typically greater than or equal to $n$. MOT is then reframed as matching these CITs to produce the final object trajectories, denoted as $G^* = \{g_1^*, g_2^*, \ldots, g_n^*\}$.

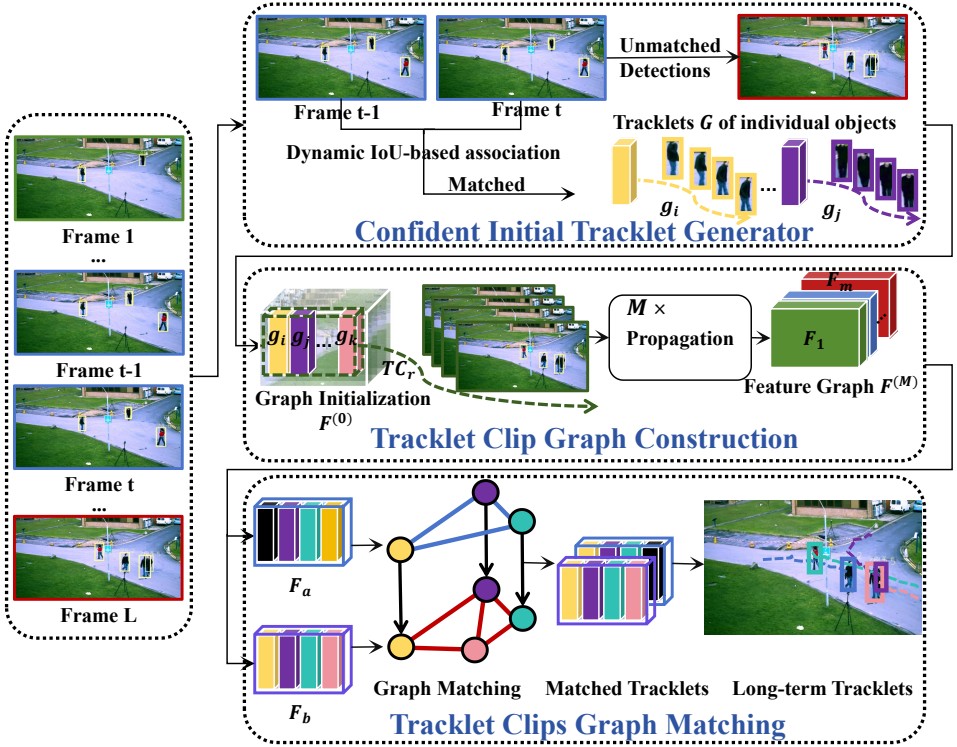

Figure 2: Overview of STAR. STAR consists of three essential components. CITG generates reliable initial tracklets $G$. The TCGC produces TCs and constructs robust spatial-temporal feature graphs $F^{(M)}$, where each object in a TC is represented as a vertex, with black cuboids serving as empty placeholders in $F_a$. TCGM facilitates efficient and accurate matching of object vertices across different TCs, ensuring robust associations.

## 3.2 Confident Initial Tracklet Generator (CITG)

A critical component of STAR is the CITG, which employs a strict association strategy. Tracklets are initialized based on object detections in the first frame and iteratively refined by calculating the IoU values between consecutive frames. If a detection matches multiple tracklets, none are selected. By prioritizing detections with higher IoU values, the generated tracklets correspond to the same object across frames. A tracklet is considered terminated if it remains unmatched for three consecutive frames, ensuring that only reliable and consistent tracklets are retained. To enhance generality while maintaining reliable associations, the IoU threshold is dynamically adjusted based on factors such as

object velocity, detection box size, and inter-frame time intervals. The reliably generated Confident Initial Tracklets (CITs) provide a solid foundation for the subsequent stages of TCGC and TCGM. Additional details can be found in Section A.2.

## 3.3 Tracklet Clip Graph Construction (TCGC)

Since CITs $G = \{g_1, g_2, \ldots, g_k\}$ have been obtained, two tasks will then be performed: generating Tracklet Clips (TCs) and constructing the TC graph, as shown in Figure 3.

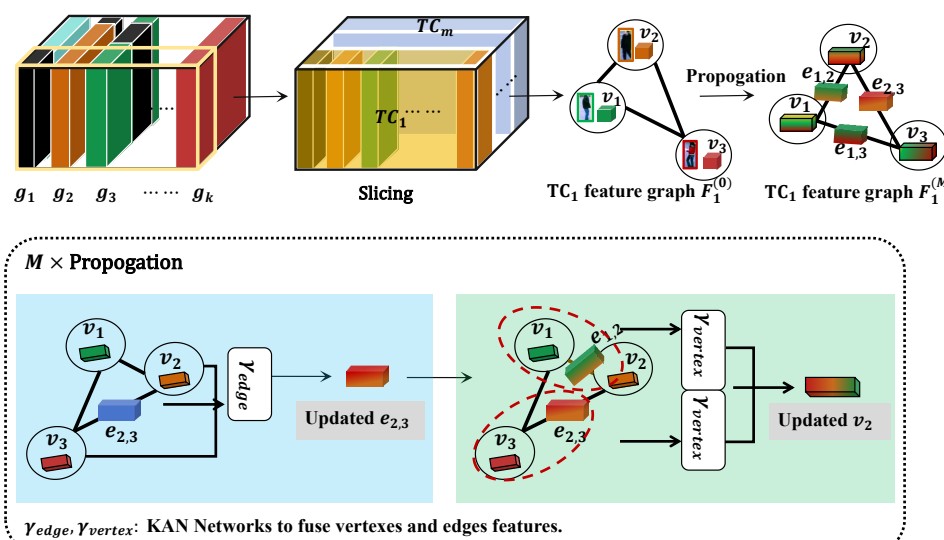

Figure 3: Overview of TCGC.

**Generating Tracklet Clips.** We arrange $G$ by aligning in time and then cut them from time dimension, into fixed length Tracklet Clips (TC). Note that each TC contains trajectory segments of the objects detected within that short time interval. For object that haven't appeared in a frame in the TC, we use an empty placeholder which is represent by black cuboid in Figure 3. We denote the generated TCs by $TC = \{TC_1, TC_2, \ldots, TC_m\}$, and the interval length of each TC is $N$ consecutive frames. A TC contains at most $k$ objects. For each tracklet segment of object, we apply weighted pooling to the position and appearance features based on confidence scores, resulting in the aggregated feature $g_l^i = \{pos_l^i, \mathbf{f}_l^i\}$ for the $l$th object in the $i$th TC.

**Constructing TC Graph.** Each TC encodes the temporal features of the concurrently appeared objects during that interval. We then construct a graph for each TC, to further model the spatial, appearance, and topology features of the objects in TC. It involves two steps: **initialization** and **propagation**. Initialization builds a graph which treats the objects in the TC as vertices, and builds edges by distances and angles. Propagation refers to the message-passing process that extracts high-order topological features. The appearance feature is also considered, and the two types of features are concatenated. Since every TC is constructed using the same method, the superscript of TC is omitted. The constructed graph of each TC successfully encode the temporal, spatial and appearance features of the objects in the TC, making each object have more distinctive feature.

### 3.3.1 Initialization of a TC Graph

Appearance and topology features are utilized, which ensure stability under varying viewpoints. Due to dimensional differences between the topology and appearance features, we divide the TC feature graph $\mathbf{F} = (V, E)$ into two parts: $\mathbf{F}_{\text{app}}$ and $\mathbf{F}_{\text{topo}}$. Each object is treated as a vertex. Let $\mathbf{n}_k$ and $\tilde{\mathbf{n}}_k$ represent the topology and appearance features of the $k$th vertex, and $\mathbf{e}_{j,k}$ and $\tilde{\mathbf{e}}_{j,k}$ represent the topology and appearance features of the edge $(j, k)$ respectively. Finally, we combine these two feature graphs as $\mathbf{F} = \mathbf{F}_{\text{app}} \cup \mathbf{F}_{\text{topo}}$, where $\mathbf{F}_{\text{topo}} = (\mathbf{n}, \mathbf{e})$ and $\mathbf{F}_{\text{app}} = (\tilde{\mathbf{n}}, \tilde{\mathbf{e}})$.

Vertex topology features are constructed using normalized lengths, angles, and positions relative to neighboring objects. We define the neighborhood set for vertex $k$ as $\mathcal{N}_k = \{g_k \mid g_k \in$

$TC$ and $\mathrm{dist}(g_j, g_k) \leq r \times \min(h_0, w_0)\}$, where $r < 1$ is a constant, and $h_0$ and $w_0$ are the height and width of the input image, respectively. The function $\mathrm{dist}(\cdot)$ represents the Euclidean distance between vertices. Then, the vertex topology features are as follows.

$$\mathbf{n}_j^{(0)} = h_{\mathrm{topo}}\left(\mathrm{pos}_j \,\|\, \mathbf{l}_j \,\|\, \theta_j\right), \quad \text{where}$$
$$\mathrm{pos}_j = \left[\frac{x_j}{w_0}, \frac{y_j}{h_0}, \frac{w_j}{w_0}, \frac{h_j}{h_0}\right], \quad \mathbf{l}_j = \left[\frac{\mathrm{dist}(g_j, g_k)}{\max(h_0, w_0)}\right] \quad g_k \in \mathcal{N}_j. \tag{1}$$

$\|$ denotes concatenation, $(x_j, y_j)$ are the bounding box center coordinates, and $(w_j, h_j)$ are the bounding box dimensions. $\theta_j$ represents the angle between adjacent neighborhood objects. We concatenate position, distance, and angle information and pass it through $h_{\mathrm{topo}}$, a Kolmogorov-Arnold Network (KAN) [43]. Edges are established between each vertex and its neighbors. The edge features are initialized based on positional and feature similarities.

$$\mathbf{e}_{j,k}^{(0)} = g_{\mathrm{topo}}\left(\left[\frac{x_j - x_k}{w_0}, \frac{y_j - y_k}{h_0}, \mathrm{IOU}\left(\mathrm{pos}_j, \mathrm{pos}_k\right), \log\left(\frac{w_j}{w_k}\right), \log\left(\frac{h_j}{h_k}\right), \cos(\mathbf{f}_j, \mathbf{f}_k)\right]\right). \tag{2}$$

The superscript $(0)$ indicates the initial state. $\cos(\cdot)$ denotes cosine similarity, and the appearance features $\mathbf{f}_j \in \mathbb{R}^{512}$ are obtained from a Re-ID module [44]. The construction of $\mathbf{F}_{\mathrm{app}}$ follows a similar approach of $\mathbf{F}_{\mathrm{topo}}$, but using object's image appearance features. To balance the dimensions of topology and appearance features, we use a three-layer KAN to reduce the dimension for vertex features in $\mathbf{F}_{\mathrm{app}}$ from 512 to 128.

### 3.3.2 Propagation

High-order features are obtained through iterative message passing. This process is executed for $M$ iterations, with messages passed from vertices to edges and then from edges to vertices. The features of two connected vertices, $\mathbf{n}_j$ and $\mathbf{n}_k$, are first fused with the corresponding edge feature $\mathbf{e}_{j,k}$. Then, each vertex $\mathbf{n}_j$ aggregates messages from its neighboring edges and incident vertices.

$$(v \rightarrow e) : \mathbf{e}_{j,k}^{(m)} = \mathbf{e}_{j,k}^{(m-1)} + \gamma_{\mathrm{edge}}\left(\left(\mathbf{n}_j^{(m-1)} + \mathbf{n}_k^{(m-1)}\right) \| \mathrm{KAN}\left(\mathbf{e}_{j,k}^{(m-1)}\right)\right)$$
$$(e \rightarrow v) : \mathbf{n}_j^{(m)} = \eta^{(Q)}\left(\mathbf{n}_j^{(m-1)}, \oplus_{\mathbf{n}_k \in \mathcal{N}_j} \gamma_{\mathrm{vertex}}\left(\mathbf{n}_j^{(m-1)}, \mathbf{e}_{j,k}^{(m)}\right)\right) \tag{3}$$

Aggregating vertex features and neighboring edge features through three levels of fusion. (1) Fusion of Neighboring Vertex and Edge Features using the aggregation function $\gamma_{\mathrm{vertex}}$, which employs a KAN and a ReLU activation function.(2) Aggregation of Neighboring Features using a permutation-invariant and differentiable aggregation function $\oplus$. (3) Final Vertex Feature Update using the function $\eta^m$.The update step $\eta(\cdot)$ is based on the message normalization proposed in [45].

$$\eta\left(\mathbf{n}_i, \mathbf{m}_i\right) = \mathrm{KAN}\left(\mathbf{n}_i + s \cdot \|\mathbf{n}_i\|_2 \frac{\mathbf{m}_i}{\|\mathbf{m}_i\|_2}\right) \tag{4}$$

where $s$ is a learnable factor, and $\|\cdot\|_2$ denotes the L2 norm. This step ensures that updated vertex features remain well-scaled and balanced. The same propagation process is applied to $\mathbf{F}_{\mathrm{app}}$, resulting in aggregated high-order features $\mathbf{n}_i^{(M)}$ and $\tilde{\mathbf{n}}_i^{(M)}$. Finally, the vertex feature $v_i$ of the graph $\mathbf{F}$ is obtained by concatenating these features. This process ensures that the final vertex features encode rich spatial-temporal information, enabling robust associations in subsequent tasks.

### 3.4 Tracklet Clip Graph Matching (TCGM)

After each TC graph has been constructed. The next step is to perform matching between different TCs. Our approach TCGM involves inputting the existing graph (including both vertex and edge features) into a differentiable graph matching layer [32], which produces the matching output. The core component of Tracklet Clip Graph Matching (TCGM) is the **differentiable graph matching layer**. This layer optimizes the matching process between detections and tracklets by solving a Quadratic Programming (QP) problem [32]. The result is an optimal matching score vector $x$, reshaped into a matrix form $\mathbf{X} \in \mathbb{R}^{N_1 \times N_2}$ to generate the matching score map. To ensure effective training, the gradients of the graph matching layer are computed using the KKT conditions, aided by the implicit function theorem [32]. TCGM leverages spatial-temporal information to establish

robust associations between tracklets, significantly reducing identity switches caused by occlusions. By modeling interactions between tracklets within a learnable and differentiable framework, TCGM improves the accuracy of the association stage, resulting in precise and reliable trajectory generation for MOT tasks.

**Complexity Analysis.** Previous graph matching methods construct graphs for individual objects in each frame, resulting in $n_{\text{graph}} = n \times \mathcal{L}$ graphs, where $n$ is the number of objects and $\mathcal{L}$ is the number of frames. The complexity of graph matching algorithms typically ranges from $O(n_{\text{graph}}^3)$ to $O(n_{\text{graph}}^2 \log n_{\text{graph}})$. By proposing TC graphs, we reduce the number of graphs to approximately $\frac{2}{N \times k}$ of the original count, remarkably improving the efficiency of graph matching.

## 3.5 Loss

To train the differentiable graph matching layer, we use a weighted binary cross-entropy loss function.

$$\mathcal{L}_{\text{track}} = \frac{-1}{N_1 N_2} \sum_{i=1}^{N_1} \sum_{j=1}^{N_2} (N_2 - 1) \, y_{j,k} \log\left(\hat{y}_{j,k}\right) + (1 - y_{j,k}) \log\left(1 - \hat{y}_{j,k}\right), \tag{5}$$

where $\hat{y}_{j,k}$ is the predicted matching score between tracklet $g_j$ and tracklet $g_k$, and $y_{j,k}$ is the ground truth indicating whether tracklet $g_j$ belongs to tracklet $g_k$. $(N_2 - 1)$ is the weight used to balance the contributions of positive and negative samples to the loss. Due to the QP-based formulation of graph matching, the resulting score map $\mathbf{X}$ tends to have a relatively smooth distribution. To sharpen this distribution and improve the focus on high-confidence matches, we apply a softmax function with temperature $\tau$.

$$\hat{y}_{j,k} = \text{Softmax}\left(x_{j,k}, \tau\right) = \frac{\exp\left(x_{j,k}/\tau\right)}{\sum_{j=1}^{N_2} \exp\left(x_{j,k}/\tau\right)}, \tag{6}$$

where $x_{j,k}$ is the original matching score from the score map $\mathbf{X}$, and $\tau$ is the inverse temperature parameter. A smaller $\tau$ sharpens the distribution, while a larger $\tau$ smooths it. Moreover, the method is applicable in both public and private detection settings. For public detection, the provided detections are directly used for tracking. For private detection, the training data is utilized to fine-tune the private detector. The total loss includes both detection and tracklet matching loss.

$$\mathcal{L} = \mathcal{L}_{\text{detect}} + \mathcal{L}_{\text{track}}, \tag{7}$$

where $\mathcal{L}_{\text{detect}}$ denotes the object detection loss, which depends on the private detector.

## 4 Experiments

### 4.1 Datasets and Metrics

**Datasets.** We select a variety of challenging benchmarks. The MOTChallenge [46, 47] datasets feature diverse scenes, viewpoints, and weather conditions. MOT17 includes 14 videos (7 for training) with three detection types: DPM [6], Faster R-CNN [7], and SDP [5]. MOT20 focuses on crowded scenes with 8 videos (4 for training and 4 for testing) that employ Faster R-CNN [7] detections. DanceTrack [48] contains 100 videos of various group dances, while VisDrone2021-MOT [49] comprises 96 sequences with around 40,000 frames across five object categories, posing challenges like occlusions and varying lighting conditions. These datasets present common issues in Multiple Object Tracking (MOT), such as frequent occlusions, irregular movements, and similar appearances, facilitating a comprehensive evaluation of STAR's robustness and generalization capabilities.

**Metrics.** The main metric for evaluating our method is the Higher Order Tracking Accuracy (HOTA) [50], which provides a balanced measure of both object detection accuracy (DetA) and association accuracy (AssA). Additionally, we also report MOTA [51], IDF1 [52], False Positives (FP), False Negatives (FN), Identity Switches (ID Sw.), and Frames Per Second (FPS) metrics.

### 4.2 Implementation Details

We employ several common data augmentation techniques, including random resize, crop, and color jitter. The input images are resized such that the shorter side is 800 pixels and the longer side is 1440

pixels. The proposed method is implemented using PyTorch. During training, $2N$ frames are sampled from each tracklet, resized to $256 \times 128$ pixels, and divided into two $N$-frame clips to enhance feature representation. The initial learning rate is set to $0.0003$ and is reduced by a factor of $0.1$ every $40$ epochs. The model is trained for 150 epochs using the Adam optimizer with a mini-batch size of 32. Additional details and discussions can be found in Section A.4.

## 4.3 Comparison with State-of-the-Art Methods

We compare STAR with numerous previous methods on the MOTChallenge [46, 47], DanceTrack [48] and VisDrone2021-MOT [49] benchmarks, as shown in Table 1, Table 2, Table 3 and Table 4, respectively. YOLOX [8] is used as the detector to ensure a fair comparison. The best results are indicated in **bold**, and the second-best results are in underline.

**MOTChallenge**. We evaluate STAR in the private detection setting and compare its performance against several state-of-the-art algorithms. The results presented in Table 1 and Table 2 indicate that STAR consistently outperforms existing methods. Our tracklet-level paradigm effectively extracts and utilizes distinctive tracklet features, achieving improvements of $1.3\%$ and $3.0\%$ in HOTA on the MOT17 and MOT20 datasets, respectively, compared to SUSHI [53].

Table 1: Performance comparison with state-of-the-art methods on the MOT17 [46] test set.

| Method | HOTA | MOTA | IDF1 | ID Sw. |
|---|---|---|---|---|
| FairMOT [10] | 59.3 | 73.7 | 72.3 | 3,303 |
| GRTU [54] | 62.0 | 74.9 | 75.0 | 1,812 |
| TLR [55] | 60.7 | 76.5 | 73.6 | 3,369 |
| MAA [15] | 62.0 | 79.4 | 75.9 | 1,452 |
| GTR [56] | 59.1 | 75.3 | 71.5 | 2,859 |
| MO3TR [57] | 60.3 | 77.6 | 72.9 | 2,847 |
| ByteTrack [41] | 63.1 | 80.3 | 77.3 | 2,196 |
| GMTracker [32] | 64.9 | 80.6 | 79.8 | 1,197 |
| FeatureSORT [58] | 64.2 | 80.6 | 76.7 | 2,637 |
| SMILEtrack [17] | 65.3 | 81.1 | 80.5 | 1,246 |
| OmniTrack [13] | 62.3 | 79.1 | 75.6 | 1,968 |
| BPMTrack [59] | 63.6 | 81.3 | 78.1 | 2,010 |
| SparseTrack [60] | 65.1 | 81.0 | 80.1 | 1,170 |
| BoostTrack++[61] | 66.6 | 80.7 | 82.2 | 1,062 |
| OccluTrack+[62] | 66.8 | 80.2 | 82.8 | **951** |
| SUSHI[53] | 66.5 | 81.1 | **83.1** | 1,149 |
| **STAR** | **67.8** | **81.9** | 80.2 | 1,057 |

Table 2: Performance comparison with state-of-the-art methods on the MOT20 [47] test set.

| Method | HOTA | MOTA | IDF1 | ID Sw. |
|---|---|---|---|---|
| FairMOT [10] | 54.6 | 61.8 | 67.3 | 5,243 |
| MAA [15] | 57.3 | 73.9 | 71.2 | 1,331 |
| ReMOT [63] | 61.2 | 77.4 | 73.1 | 1,789 |
| ByteTrack [41] | 61.3 | 77.8 | 75.2 | 1,223 |
| GMTracker [32] | 62.9 | 77.8 | 76.7 | 1,331 |
| BPMTrack [59] | 62.3 | 78.3 | 76.7 | 1,314 |
| BASE [16] | 63.5 | 78.2 | 77.6 | 984 |
| SparseTrack [60] | 63.4 | 78.1 | 77.6 | 1,120 |
| BoostTrack++[61] | 66.4 | 77.7 | 82.0 | 762 |
| OccluTrack+[62] | 66.7 | 77.7 | **82.7** | **429** |
| SUSHI[53] | 64.3 | 74.3 | 79.8 | 706 |
| **STAR** | **67.3** | **79.6** | 79.0 | 1,047 |

**DanceTrack**. The complex scenarios characterized by frequent occlusions and irregular motion present significant challenges for tracking systems. Using the same pre-trained detector, STAR shows substantial improvements over SparseTrack [60], with improvement of $0.4\%$ in HOTA, $0.5\%$ in MOTA, $0.4\%$ in AssA, and $0.2\%$ in DetA as indicated in Table 3.

Table 3: Performance comparison with state-of-the-art methods on the DanceTrack [48] test set.

| Method | HOTA | MOTA | IDF1 | AssA | DetA |
|---|---|---|---|---|---|
| CenterTrack [19] | 41.8 | 86.8 | 35.7 | 22.6 | 78.1 |
| TraDes [12] | 43.3 | 86.2 | 41.2 | 25.4 | 74.5 |
| OCSORT [33] | 55.1 | **92.0** | 54.6 | 38.3 | **80.3** |
| FairMOT [10] | 39.7 | 82.2 | 40.8 | 23.8 | 66.7 |
| QDTrack [64] | 54.2 | 87.7 | 50.4 | 36.8 | 80.1 |
| GTR [56] | 48.0 | 84.7 | 50.3 | 31.9 | 72.5 |
| ByteTrack [41] | 47.7 | 89.6 | 53.9 | 32.1 | 71.0 |
| BoT-SORT [11] | 54.7 | 91.3 | 56.0 | 37.8 | 79.6 |
| SparseTrack [60] | 55.5 | 91.3 | **58.3** | 39.1 | 78.9 |
| **STAR** | **55.9** | 91.8 | 57.9 | **39.5** | 79.1 |

These results highlight the considerable potential of our method in managing occlusion scenarios. Notably, even with a simple IoU distance association strategy, STAR achieves comparable or even superior performance relative to other methods.

**VisDrone2021-MOT**. This dataset presents even greater challenges due to frequent occlusions and varying lighting conditions. As shown in Table 4, STAR surpasses BoT-SORT by 2.9% in HOTA and 1.4% in IDF1, while outperforming OCSORT [33] by 2.4% in HOTA and 6.2% in IDF1. Furthermore, STAR demonstrates a 0.4% improvement in IDF1 over UGT while achieving a 1.6 FPS advantage, highlighting its efficiency in aerial tracking tasks.

Table 4: Performance comparison with state-of-the-art methods on the VisDrone2021-MOT [49] test set.

| Method | HOTA | MOTA | IDF1 | FN | FP | ID Sw. | FPS |
|---|---|---|---|---|---|---|---|
| DeepSORT [65] | 36.9 | 34.4 | 46.7 | 110,989 | 21,077 | 1,784 | 18.5 |
| ByteTrack [41] | 40.7 | 39.5 | 50.4 | 105,518 | 16,257 | 1,581 | **31.2** |
| BoT-SORT [11] | 42.4 | 41.7 | 56.8 | 103,505 | 14,114 | 1,430 | 25.3 |
| BiOU_Tracker [66] | 40.2 | 40.7 | 48.8 | 103,188 | 15,794 | 2,029 | 28.5 |
| MOTDT [67] | 37.0 | 35.5 | 52.6 | 106,006 | 15,385 | 2,668 | 26.2 |
| OCSORT [33] | 42.9 | 41.6 | 52.0 | 132,279 | 22,019 | 2,859 | 15.4 |
| StrongSORT [35] | 36.6 | 33.3 | 42.5 | 185,503 | 12,214 | 1,980 | 12.1 |
| UCMCTrack [14] | 37.1 | 28.1 | 38.4 | 150,590 | 9,244 | 7,231 | 17.8 |
| QDTrack [64] | 44.7 | 39.1 | 55.3 | 104,759 | 34,242 | 2,627 | 10.3 |
| OUTrack [68] | 34.0 | 35.0 | 44.5 | 115,570 | 25,276 | 3,335 | 17.4 |
| FairMOT [10] | 31.1 | 12.8 | 37.7 | 114,834 | 59,997 | 3,072 | 21.5 |
| TrackFormer [69] | 35.3 | 25.0 | 51.0 | 141,526 | 25,856 | 1,534 | 7.0 |
| SGT [24] | 43.6 | 39.2 | 54.8 | 110,652 | **7,707** | 951 | 27.9 |
| UGT [25] | **45.5** | 41.8 | 57.8 | 101,074 | 15,174 | 618 | 16.2 |
| STAR | 45.3 | **41.9** | **58.2** | **100,832** | 15,289 | **602** | 17.8 |

These results emphasize STAR's strong tracking stability and superior performance. Its effective modeling of distinctive spatial-temporal tracklet feature and the matching of TC graph enable STAR to achieve state-of-the-art results on MOTChallenge, DanceTrack, and VisDrone2021-MOT, establishing a new benchmark for multi-object tracking (MOT).

## 4.4 Ablation analysis

### 4.4.1 Effect of Tracklet Clip Graph Construction (TCGC)

Each TC consists of N consecutive frames. The analysis of the frame count (N) per TC is shown in Table 5 and Figure 4. It can be observed that $N = 6$ performs the best and $N = 4$ is the second-best. Although there was a slight improvement at $N = 6$, it was not significant. Therefore, we opted for $N = 4$ to ensure robust performance in challenging scenes.

For the sampling strategy with $N = 4$ and a stride length of $L$ in TCGC, we experimented with various stride lengths to assess their impact on model performance. The detailed experimental results are presented in Table 6. The model achieved optimal performance at a stride length of $L = 1$. This smaller stride length allowed for more effective capture of continuity and changes over time, thus enhancing the model's temporal data processing capabilities. Furthermore, comparing stride lengths of $L = 2$ and $L = 1$ with a window size of $N = 4$, we found that although both settings yielded good performance, the $L = 2$ configuration provided high accuracy and improved efficiency.

Appearance similarity serves as a baseline for tracklet association. As shown in Table 7, distinctive and robust tracklet features significantly enhance tracking performance compared to models that rely solely on IoU distance for association. While Global Average Pooling (GAP) utilizes a simpler pooling strategy, our TCGC improves IDF1 from 61.9% to 69.2%. These results clearly highlight the importance of advanced tracklet feature extraction.

### 4.4.2 Effect of STAR

We demonstrate that integrating our proposed STAR method with existing tracking approaches enhances their performance, as summarized in Table 8. The evaluated baseline methods, Deep-SORT [65], JDE [70], and CTracker [71], utilize IoU distance and frame-level features for tracklet

Table 5: Performance evaluation for various N values.

| N | mAP | Top-1 | Top-5 | Top-10 |
|---|------|-------|-------|--------|
| 2 | 91.14 | 89.52 | 98.89 | 98.89 |
| 4 | 92.52 | 91.84 | 98.89 | 98.89 |
| 5 | 92.55 | 91.92 | 98.89 | 98.89 |
| 6 | 92.60 | 91.95 | 98.89 | 98.89 |
| 7 | 92.43 | 91.36 | 98.89 | 98.89 |
| 8 | 92.04 | 90.88 | 98.89 | 98.89 |

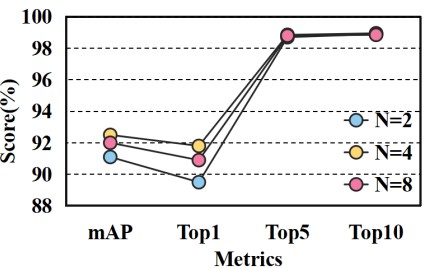

Figure 4: Effect of N in TCGC.

Table 6: Experimental results for various combinations of window size $N$ and stride length $L$.

| N | L | HOTA | MOTA | IDF1 | ID Sw. |
|---|---|------|------|------|--------|
| 4 | 1 | 67.9 | 82.1 | 80.4 | 1,023 |
| 4 | 2 | 67.8 | 81.9 | 80.2 | 1,057 |
| 4 | 3 | 67.4 | 81.5 | 79.6 | 1,129 |
| 4 | 4 | 67.0 | 81.2 | 79.5 | 1,327 |

Table 7: Effect of Different Tracklet Description Methods.

| Methods | IDF1 | HOTA | MOTA | ID Sw. |
|---------|------|------|------|--------|
| Baseline | 61.9 | 55.5 | 57.6 | 296 |
| GAP | 63.1 | 55.6 | 57.6 | 209 |
| STTA [20] | 68.6 | 58.7 | 57.7 | 133 |
| TCGC | **69.2** | **59.0** | **57.9** | **125** |

generation, with CTracker also incorporating topological information for data association. When comparing the baseline versions with their modified counterparts (DeepSORT*, JDE*, and CTracker*) that rely solely on IoU distance, we observe comparable MOTA scores but a significant drop in IDF1 (from 62.0% to 55.0%). This decline highlights IoU distance limitations in maintaining identity consistency across long trajectories and emphasizes the necessity for advanced feature extraction and data association techniques. Integrating STAR with these methods boosts performance, improving IDF1 by 3.2%, 3.5%, and 8.0% for DeepSORT, JDE, and CTracker, respectively. These results demonstrate STAR's effectiveness in enhancing identity preservation and association accuracy. In conclusion, STAR is model-agnostic and can be seamlessly integrated into various tracking frameworks.

Table 8: Performance of Adding STAR upon Existing Methods on the MOT16 Training Dataset.

| Methods | MOTA | IDF1 | HOTA | FP | FN | ID Sw. |
|---------|------|------|------|-----|-----|--------|
| Deepsort [65] | 56.9 | 62.0 | 51.3 | 13,227 | 33,454 | **932** |
| Deepsort* | 56.6 | 55.0 | 48.3 | **10,433** | 35,627 | 1,883 |
| Deepsort*+STAR | **58.8** | **65.2** | **54.3** | 13,041 | **31,105** | 1,315 |
| JDE [70] | 73.1 | 68.9 | 55.1 | 6,593 | 21,788 | 1,312 |
| JDE* | 73.0 | 61.9 | 53.6 | **6,185** | 22,296 | 1,330 |
| JDE*+STAR | **73.5** | **72.4** | **55.3** | 6,871 | **21,350** | **1,125** |
| CTracker [71] | 76.2 | 68.6 | 61.1 | **2,149** | 23,188 | 912 |
| CTracker* | 76.2 | 68.6 | 61.1 | **2,149** | 23,188 | 912 |
| CTracker*+STAR | **78.8** | **76.6** | **66.0** | 3,981 | **18,960** | **540** |

## 5 Limitation and Conclusion

Unlike previous approaches that overlook historical information and temporal continuity, STAR extracts and effective leverages distinctive tracklet features through TC to address occlusion challenges. The framework consists of three core components. CITG efficiently generates reliable CITs, TCGC produces discriminative TC feature graphs by exploring spatial-temporal information within tracklets, and TCGM uses graph matching to enhance association accuracy and improve efficiency. Together, these components produce high-integrity trajectories and achieve state-of-the-art performance across three widely used benchmarks, demonstrating STAR's effectiveness and robustness. Despite its strong performance, the method faces efficiency limitations. Future work will focus on designing an end-to-end tracking framework to further enhance STAR's robustness and applicability in real-world multi-object tracking (MOT) scenarios.

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

# A Appendix

## A.1 Overview

In the supplementary material, we primarily:

1. Provide more details of CITG in Appendix A.2.

2. Provide a overview of TCGM in Appendix A.3.

3. State more experimental details in Appendix A.4.

4. Provide additional experimental results in Appendix A.5.

## A.2 Construction of CITG

A short-term trajectory continues to be associated with the detection in the next frame until it no longer matches any detection. The Confident Initial Tracklet Generator utilizes the Intersection over Union (IoU) metric to associate objects by calculating the overlap between detection boxes. Below is the detailed process.

**Trajectory Initialization:** In the first frame of the video, each detection box is initialized as an independent short-term trajectory.

**IoU Calculation:** For each trajectory in the current frame, the IoU with all detection boxes in the subsequent frame is calculated. The IoU value, ranging from 0 to 1, indicates the degree of overlap, with higher values denoting greater overlap.

**Trajectory Matching:** Trajectories are matched to detection boxes based on IoU values. A trajectory and a detection box from the next frame are considered to belong to the same object if their IoU exceeds a specific threshold. If a detection box matches multiple trajectories, none are selected. Unmatched detection boxes are initialized as new trajectories.

**Dynamic Adjustment of IoU Threshold:** To accommodate varying motion characteristics of targets, a dynamic adjustment method for the IoU threshold is employed. This method considers multiple factors, including the target's motion speed ($v$), detection box size ($A$), and time interval between frames ($\Delta t$). These factors dynamically influence the IoU threshold.

**Trajectory Update and Termination:** Matched detection boxes are added to corresponding trajectories, updating their state (such as location and timestamp). Trajectories that remain unmatched for more than three consecutive frames are terminated. Unmatched detection boxes initiate new trajectories.

**Output Short-Term Trajectories:** The aforementioned steps generate a set of short-term trajectories that capture the preliminary motion trajectories of all targets in the video. These trajectories form the basis for further trajectory association and long-term trajectory generation.

## A.3 Overview of TCGM

## A.4 More Implementation Details

We train our model on 24 NVIDIA RTX 2080Ti GPUs. During testing, the entire tracklet is used as input, with every N=4 frames treated as a clip.

**MOTChallenge.** In the MOT16 dataset [46], only objects with a visible ratio greater than 0.3 are selected, resulting in 517 training identities, 438 gallery identities, and 429 query identities. The total number of training videos is 2,065, and there are 2,173 testing videos, with each ground truth trajectory divided into four variable-length tracklets.

**VisDrone2021-MOT.** The VisDrone2021-MOT-train set, which consists of 56 sequences, is used for training, while the VisDrone2021-MOT-test-dev set, containing 17 sequences, is used for testing. During evaluation, a single tracklet per identity serves as the query, with the remaining tracklets in the gallery.

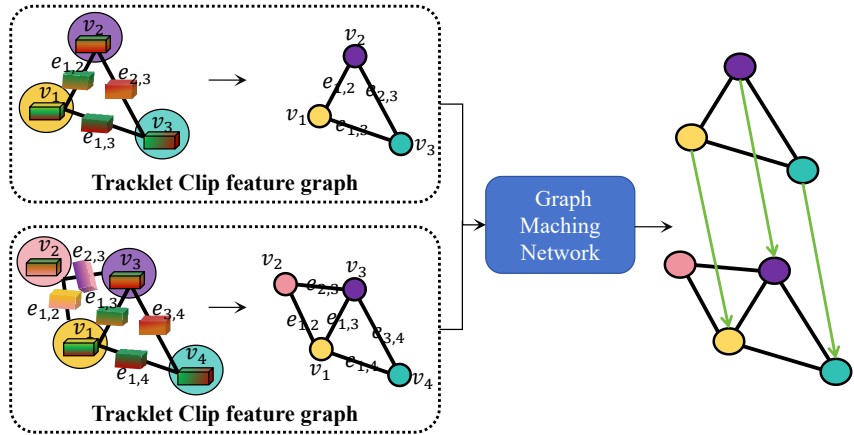

Figure 5: Overview of TCGM.

**UAVDT.** STAR is also evaluated on the UAVDT dataset [72], which provides diverse scenarios and challenges. The UAVDT dataset contains 50 sequences with over 80,000 frames, capturing various situations that present frequent occlusions and significant camera motion.

## A.5 More Comparisons with State-of-the-Art Methods

We compare the proposed STAR method with traditional **TBD** approaches using Tracktor to refine detections in the public detection setting. As shown in Table 9, STAR outperforms existing methods on most evaluation metrics, achieving significant improvements in MOTA, IDF1, and HOTA across the MOT15, MOT16, MOT17, and MOT20 datasets. The tracklet-level approach of STAR integrates tracklet information during feature extraction and data association, which facilitates more robust tracking.

Table 9: Performance comparison of Public Detection with state-of-the-art methods on the MOTChallenge test set. The best results are indicated in **bold**, and the second-best results are in underline.

| Datasets | Methods | HOTA | MOTA | IDF1 | FP | FN | ID Sw. |
|---|---|---|---|---|---|---|---|
| MOT15 | Tracktor [73] | 37.6 | 46.6 | 47.6 | **4,624** | 26,896 | 1,280 |
| | LFP [74] | 43.8 | 47.2 | 57.6 | 7,635 | 24,277 | 554 |
| | MPNTrack [75] | 45.0 | 51.5 | 58.6 | 7,620 | 21,780 | **375** |
| | STAR | **45.8** | **55.8** | **59.9** | 5,983 | **20,700** | 475 |
| MOT16 | Tracktor [73] | 44.6 | 56.2 | 54.9 | 2,394 | 76,844 | 617 |
| | GSM [22] | 45.9 | 57.0 | 58.2 | 4,332 | 73,543 | 475 |
| | LFP [74] | 49.6 | 57.5 | 64.1 | 4,249 | 72,868 | 335 |
| | MPNTrack [75] | 48.9 | 58.6 | 61.7 | 4,949 | 70,252 | 354 |
| | LPC [76] | **51.7** | 58.8 | **67.6** | 6,167 | 68,432 | 435 |
| | MO3TR [57] | 50.3 | **64.2** | 60.6 | 7,620 | **56,761** | 929 |
| | TMOH [77] | 50.7 | 63.2 | 63.5 | 3,122 | 63,376 | 635 |
| | GMTracker [32] | 48.9 | 55.9 | 63.9 | **2,371** | 77,545 | 531 |
| | STAR | 51.2 | 64.1 | 66.5 | 2,427 | 62,377 | 511 |
| MOT17 | Tracktor [73] | 44.8 | 56.3 | 55.1 | 8,866 | 235,449 | 1,987 |
| | GSM [22] | 45.7 | 56.4 | 57.8 | 14,379 | 230,174 | 1,485 |
| | LFP [74] | 50.7 | 58.2 | 65.2 | 16,850 | 217,944 | **1,022** |
| | MPNTrack [75] | 49.0 | 58.8 | 61.7 | 17,413 | 213,594 | 1,185 |
| | LPC [76] | 51.5 | 59.0 | **66.8** | 23,102 | 206,948 | 1,122 |
| | MO3TR [57] | 49.6 | 63.2 | 60.2 | 21,966 | **182,860** | 2,841 |
| | TMOH [77] | 50.4 | 62.1 | 62.8 | 10,951 | 201,135 | 1,897 |
| | GMTracker [32] | 49.1 | 56.2 | 63.8 | **8,719** | 236,541 | 1,778 |
| | STAR | **52.5** | **64.2** | 66.5 | 8,971 | 190,636 | 1,994 |
| MOT20 | Tracktor [73] | 42.1 | 52.6 | 52.7 | **6,930** | 236,680 | **1,648** |
| | TMOH [77] | 48.9 | 60.1 | 61.2 | 8,043 | 165,899 | 2,342 |
| | STAR | **52.9** | **64.1** | **66.5** | 39,357 | **143,583** | 2,825 |

For the UAVDT dataset, 40 sequences are randomly selected for training, while 10 sequences are designated for testing. As summarized in Table 10, STAR demonstrates superior performance on the UAVDT dataset, achieving HOTA, MOTA, and IDF1 scores of 63.4%, 71.4%, and 80.3%, respectively, surpassing all other methods. STAR outperforms BoT-SORT [11] by 2.0% in HOTA and 3.7% in MOTA, highlighting its effective modeling of topological relationships. Additionally, STAR exceeds UGT [25], the best-performing method on UAVDT, by 1.4 FPS in terms of computational efficiency.

Table 10: Performance comparison with state-of-the-art methods on the UAVDT [72] test set. The best results are indicated in **bold**, and the second-best results are in underline.

| Method | HOTA | MOTA | IDF1 | FN | FP | ID Sw. | FPS |
|---|---|---|---|---|---|---|---|
| DeepSORT [65] | 62.0 | 68.5 | 78.6 | 20,035 | 4,008 | **61** | 20.3 |
| ByteTrack [41] | 62.2 | 68.8 | 78.8 | 20,010 | 3,796 | 102 | 32.1 |
| BoT-SORT [11] | 61.4 | 67.7 | 78.5 | 20,296 | 4,323 | 64 | 25.0 |
| BiOU_Tracker [66] | 62.9 | 70.3 | 79.6 | 17,405 | 5,224 | 79 | **38.1** |
| MOTDT [67] | 61.8 | 66.5 | 77.8 | 17,760 | 5,824 | 76 | 22.4 |
| OCSORT [33] | 59.8 | 69.5 | 74.5 | 18,246 | 6,480 | 249 | 13.3 |
| StrongSORT [35] | 58.3 | 49.3 | 72.6 | **10,606** | 28,032 | 134 | 10.6 |
| UCMCTTrack [14] | 54.1 | 61.0 | 65.9 | 25,888 | **2,482** | 984 | 15.4 |
| QDTrack [64] | 61.2 | 70.3 | 76.1 | 17,304 | 6,420 | 92 | 13.4 |
| OUTrack [68] | 58.6 | 64.3 | 66.5 | 18,193 | 7,826 | 240 | 18.2 |
| FairMOT [10] | 49.1 | 51.2 | 66.5 | 33,102 | 4,136 | 110 | 25.5 |
| TrackFormer [69] | 43.2 | 37.9 | 53.3 | 45,197 | 5,582 | 680 | 7.94 |
| SGT [24] | 58.2 | 57.8 | 77.0 | 11,658 | 20,598 | 102 | 30.0 |
| UGT [25] | **63.6** | **71.6** | 80.0 | 17,285 | 4,357 | 67 | 18.3 |
| STAR | 63.4 | 71.4 | **80.3** | 17,462 | 4,389 | 74 | 19.7 |

