# OpenReview forum: "STAR: Spatial-Temporal Tracklet Matching for Multi-Object Tracking"
_NeurIPS.cc/2025/Conference — NeurIPS 2025 poster_

### Official Review · Reviewer_5M6K · 2025-07-01

**Clarity:** 3
**Significance:** 2
**Originality:** 2
**Rating:** 3
**Confidence:** 4

**Summary:**

The paper proposes an MOT method under the tracking-by-detection framework.  It uses graph representation learning to explore spatial-temporal information within tracklets. And it uses tracklet clip-level graph matching to gain robustness to occlusions and ID-switches. It does experiments on three widely used benchmarks to verify its effectiveness.

**Questions:**

1)	It is suggested to resummarize its contribution to highlight its difference with existing tracklet graph matching-based methods works.
2)	Unclear technical and experiment details, to see the weakness.

**Ethical Concerns:**

["NO or VERY MINOR ethics concerns only"]

**Final Justification:**

I carefully read the rebuttal and other reviewers’ comments. I will be happy to follow other reviewers’ opinions and have no counterview if this paper is accepted. However, I still consider its claim on contribution/novelty is incorrect. So my final rating is borderline.

[24,a] both simultaneously introduce temporal information and topological relations.
[24] includes an association graph to leverage the topological relationships both spatially and temporally.
[a] considers temporal information and topological relations by defining a learnable interaction representation which captures the latent representations of all other tracklets whose lifespan overlaps with the temporal range of the target tracklet.

In fact, a MOT work will not consider topological relations alone but not consider temporal information.

It claims “Previous graph matching methods construct graphs for individual objects in each frame, resulting in  n*L graphs”.
Note that many exiting graph matching methods construct graphs for tracklets, i.e. making associations between tracklets (the same as ‘tracklet clip’ in the paper), but NOT for individual objects in each frame. So their numbers of graphs for matching is similar to STAR. Both STAR and previous methods need initial tracklet generator, so computation complexity of this initialization part is also the same.

**Limitations:**

yes.

**Paper Formatting Concerns:**

No.

**Quality:**

2

**Strengths And Weaknesses:**

Strength
1）	Its motivation is plausible and it describes clearly different modules of the method like graph construction and propagation.

2）	Ablation studies on the tracklet clip graph construction and the STAR module are provided.

3）	It selects a variety of challenging benchmarks to verify its effectiveness.

4）	Supplementary material provides more comparisons on MOT and UAVDT datasets.

Weakness：
--Overall, the idea is not new in MOT domain which solves viewpoint variations and occlusions by exploiting spatial-temporal information.
Formulating the data association problem as a graph matching task is common in MOT. And the hierarchical association strategy from detections to tracklet clips is also commonly used. It is suggested to highlight the major differences between the existing tracklet graph matching-based methods and the proposed TCGM.

--Some technique details:
For object that haven’t appeared in a frame in the TC, they use an empty placeholder. How the appearance and position features and the topological features of such dummy nodes are computed? How to process them in the following graph matching.

They use a weighted binary cross-entropy loss function for the differentiable graph matching. Though it refers to [31], some explanations about KKT conditions and the implicit function theorem can be provided in supplementary material.

--Unclear experiment details:
-In the ablation study on the effect of STAR, the experiment configurations are unclear. Adding STAR as a postprocessing step to an existing method seems sure to bring improvement.

-How it determines to create a new trajectory or terminate an existing trajectory with the object disappearing for a long time.

-- Since the method is verified in both public and private detection settings, it is suggested to list the two types separately in experiment results. Quality of detections greatly affects the tracking performance in the TBD framework.

----The whole pipeline as in Fig. 2 is somewhat complex, which may hinder the proposed method from real application. Efficiency is an unavoidable problem in MOT, though this problem is briefly discussed in the limitation section.

---

> ### Author Rebuttal · Authors · 2025-07-31
>
> Dear Reviewer 5M6K,
>
> Thank you for the detailed review and invaluable comments. We address your concerns point by point. Feel free to ask follow-up questions if something remains unclear.
>
> ---
>
> **Q1:** It is suggested to resummarize its contribution to highlight its difference with existing tracklet graph matching-based methods works.
>
>
> **Respond to Q1:**
> Thank you very much for the valuable suggestions. We have summarized the contribution of the STAR method again and clearly highlighted its differences from the existing tracklet graph matching-based methods. Firstly, unlike traditional graph-based MOT methods that consider each target independently, STAR considers not only the features of individual targets but also the interactions between targets (i.e., the topological relationships between targets). By establishing these relationships, STAR not only enhances the distinctiveness of individual target characteristics but also effectively addresses dynamic occlusion issues using the short-term stability in relationships between targets. This approach significantly improves tracking continuity and accuracy in complex environments.
>
> Regarding the innovation in data association, STAR employs a graph matching approach where each target's tracklet segment is treated as a node in the graph. This method, compared to traditional strategies that consider each individual target as a node, greatly reduces the number of graph matching operations, thus significantly improving processing efficiency. In specific experimental tests, this technique not only drastically reduces the computational load but also enhances the smoothness and accuracy of tracking.
>
> ---
>
> **Q2:** Some technique details: For object that haven’t appeared in a frame in the TC, they use an empty placeholder. How the appearance and position features and the topological features of such dummy nodes are computed? How to process them in the following graph matching.
>
> **Respond to Q2:**
> For object that haven’t appeared in a frame in the TC, we use an empty placeholder. Then the initial features are initialized with the average of its adjacent non-empty frames.
>
> After initialization, each tracklet segment is treated as a node, and the features of these nodes are iteratively updated through TCGC. Ultimately, $m $ feature graphs are obtained.
> By performing graph matching on these feature graphs, the matching between tracklet segments is achieved.
>
> The ablation study analyzing the impact of STAR was conducted on the MOT16 dataset. We demonstrated that adding STAR to existing methods can further enhance their performance. Compared to Deepsort, JDE, and CTracker, CTracker*/Deepsort*/JDE* indicate that trajectories are generated using only the Intersection over Union (IoU) distance. The introduction of the STAR module consistently improves the performance of existing methods. The experiments indicate that STAR is a model-independent, universal method that can be seamlessly integrated into existing technologies, providing continuous performance optimization.

---

> ### Comment · Reviewer_5M6K · 2025-08-04
>
> THank the authors for detailed rebuttal. My concerns about technique details are solved.
>
> It is said STAR’s difference from existing methods lies in considering interactions between targets (i.e., the topological relationships between targets). In fact, modeling topological relationships for robust tracking has been considered in many existing MOT works, just listing 3 examples:
>
> [24]Learning a Robust Topological Relationship for Online Multiobject Tracking in UAV Scenarios
>
> [a]Probabilistic Tracklet Scoring and Inpainting for Multiple Object Tracking, CVPR2021
>
> [b]Learnable Graph Matching: Incorporating Graph Partitioning with Deep Feature Learning for Multiple Object Tracking
>
> The above works use neighborhood and topological relationships either spatially or both spatially and temporally. Some methods use the same features (i.e., relative distances and relative angles between the target and its neighbors ) as STAR to represent topological relationships, while some use other interaction or geometric features.
> Broadly speaking, the idea is widely accepted in the MOT domain that effective modeling of interactions will lead to better tracking quality as the motion of each target may be affected by the behaviour of its neighbors in the scene.
>
> As another innovation in data association, STAR treats each target's tracklet segment as a node in the graph. This is also widely adopted in tracklet graph-matching based methods.

---

> > ### Author Response · Authors · 2025-08-05
> > **Rebuttal by Authors**
> >
> > Dear Reviewer 5M6K,
> >
> > Thank you for your response. We have carefully read the papers you mentioned and realize that our previous description may not have been clear enough. To help you better understand our work, we will explain further. Feel free to ask follow-up questions if something remains unclear.
> >
> > As you mentioned, some papers introduce features based on topological relationships [24], some incorporate temporal information [a], and others use graph matching methods [b]. These papers have demonstrated that introducing these methods individually can, to some extent, improve MOT performance. However, it remains unclear whether the simultaneous introduction of temporal information and topological relations (relative positional relationships) can further enhance performance, how to balance these features, and how to overcome the efficiency bottleneck of graph matching methods. These issues have not been addressed simultaneously in previous works.
> >
> > In our method, we introduce both temporal information and topological relationship features (relative positional relationships), and innovatively propose the concept of a "Tracklet Clip."  A "Tracklet Clip" refers to a segment of confidently associated trajectories of multiple objects, capturing not only the appearance but also spatial and temporal features of an object, making it more distinctive in the feature space. Additionally, we further enhance the features of TCs by integrating topology information and high-order features through graph neural networks. We represent a segment of the trajectory’s features at one point and innovatively construct these "Tracklet Clips" into a graph.
> >
> > Simply put, if the feature graphs of previous methods are seen as rectangles, our method represents them as cuboids (as shown in Figure 3), which is a novel approach not found in any prior articles. Thus, our graphs contain richer feature information and also significantly reduce the number of graphs needed for graph matching, substantially shortening the time required for graph matching. Previous graph matching methods construct graphs for individual objects in each frame, resulting in $ n_{\text{graph}} = n \times \mathcal{L} $ graphs, where $  n$  is the number of objects and $  \mathcal{L} $  is the number of frames. The complexity of graph matching algorithms typically ranges from $ O(n_{\text{graph}}^3)$  to $  O(n_{\text{graph}}^2 \log n_{\text{graph}}) $ . By proposing TC graphs, we reduce the number of graphs to approximately$  \frac{2}{N \times k} $  of the original count, remarkably improving the efficiency of graph matching.

---

### Official Review · Reviewer_pdkw · 2025-07-02

**Clarity:** 3
**Significance:** 2
**Originality:** 2
**Rating:** 4
**Confidence:** 4

**Summary:**

The paper introduces a Spatial-Temporal Tracklet Graph Matching method for multiple object tracking. To address occlusion issues, trajectory graphs are built by combining tracklet clips and graph matching. However, there are some concerns regarding motivation and performance.

**Questions:**

1. Clarify how the proposed methods address the conclusion issues
2. Include more recent state-of-the-art works
3. Provide the ablation study on the IoU-based Association

**Ethical Concerns:**

["NO or VERY MINOR ethics concerns only"]

**Final Justification:**

The comparison is comprehensive and improved, and the proposed method shows some novelty. I increased the rating.

**Limitations:**

No, how the proposed methods address the conclusion issues.

**Quality:**

2

**Strengths And Weaknesses:**

Strengths
* The evaluation is comprehensive and conducted on several datasets.
* The presentation is clear and easy to follow.

Weaknesses
* The Confident Initial Tracklet Generator heavily relies on the Dynamic IoU-based association. However, IoU-based Associations are not strictly reliable and generate wrong tracklets, especially after occlusion. Consequently, the constructed graph can be polluted, which weakens the purpose of the graph construction. Meanwhile, the ablation study should also show the results where the tracker solely relies on the IoU-based association and the effects of the association (using association results and GTs on the MOT-17 validation set).
* The STAR aims to address the occlusion issues. However, the methods do not provide information or explanations indicating how each implementation contributes to the motivation to address the occurrence.
* In the literature review and benchmarks, some state-of-the-art methods are missing, such as BoosTrack++ and OccluTrack. Meanwhile, the performance on the MOT 17 and MOT 20 datasets is not comparable with some of those methods. In addition, HOTA should be included in the benchmark, which is the most important metric for evaluating tracking performance.
1. Stanojević, V., & Todorović, B. (2024). BoostTrack++: using tracklet information to detect more objects in multiple object tracking. arXiv preprint arXiv:2408.13003.
2. Gao, J., Wang, Y., Yap, K. H., Garg, K., & Han, B. S. (2025). Occlutrack: Rethinking awareness of occlusion for enhancing multiple pedestrian tracking. IEEE Transactions on Intelligent Transportation Systems.

---

> ### Author Rebuttal · Authors · 2025-07-31
>
> Dear Reviewer pdkw,
>
> Thank you for the detailed review and invaluable comments. We address your concerns point by point and invite you to ask any follow-up questions if anything remains unclear.
>
> ---
>
> **Q1:** The STAR aims to address the occlusion issues. However, the methods do not provide information or explanations indicating how each implementation contributes to the motivation to address the occurrence.
>
> **Response to Q1:**
>
> Thank you for your detailed review. With respect to the mechanism of STAR in managing occlusions, we have enhanced feature discernibility in occluded scenarios by integrating both temporal information and topological relationships into our system. Let me further clarify the application of these methods:
>
> **Temporal Information**: By generating tracklets and extracting features over time, STAR enables the system to infer the characteristics of a object even when direct observation is momentarily obstructed. For instance, in a dense crowd, should an individual be temporarily hidden by others, the system continues to project their likely whereabouts until they become visible again.
>
> **Topological Relationships**: The spatial interrelations among objects serve as another robust strategy to address occlusions. In scenarios like vehicle tracking, the relative positions of multiple objects typically exhibit short-term consistency. Leveraging this attribute, the STAR system preserves the integrity of the topological structure among objects, thereby facilitating the recognition and tracking of occluded objects.
>
> To validate the efficacy of these approaches, rigorous testing was conducted using the dancetrack dataset, known for its frequent occlusions. The outcomes affirm that the fusion of temporal and topological information elevates the HOTA to 55.9\%. This underscores the substantial improvement in handling occlusions, attesting to the robustness of our method.
>
> ---
>
> **Q2:** In the literature review and benchmarks, some state-of-the-art methods are missing, such as BoosTrack++ and OccluTrack. Moreover, the performance on the MOT 17 and MOT 20 datasets is not comparable with some of those methods. Additionally, HOTA should be included in the benchmark, which is the most important metric for evaluating tracking performance.
>
> **Response to Q2:**
> We acknowledge that our literature review did not include BoosTrack++ and OccluTrack. However, please understand that there are numerous studies related to MOT, and we have tried our best to keep our data fair and reliable .
>
> To address these issues, we have included comparisons with these methods on the MOT 17 and MOT 20 datasets, and have introduced the HOTA evaluation metric. This will help in more comprehensively and accurately evaluating and demonstrating the performance of our model. It can be observed that our HOTA and MOTA scores are comparatively higher, while our IDF1 and ID Sw. scores are somewhat lower. Despite this, these outcomes clearly demonstrate the effectiveness of our method.
>
> | **Datasets** | **Methods**   | **HOTA** | **MOTA** | **IDF1** | **ID Sw.** |
> |--------------|---------------|----------|----------|----------|------------|
> | **MOT17**    | BoostTrack++  | 66.6     | 80.7     | 82.2     | 1,062      |
> |              | OccluTrack+   | 66.8     | 80.2     | **82.8**     | **951**        |
> |              | STAR          | **67.8** | **81.9** | 80.2     | 1,057      |
> | **MOT20**    | BoostTrack++  | 66.4     | 77.7     | 82.0     | 762        |
> |              | OccluTrack+   | 66.7     | 77.7     | **82.7**     | **429**        |
> |              | STAR          | **67.3** | **79.6** | 79.0     | 1,047      |
>
>
> ---
>
> **Q3:** Provide the ablation study on the IoU-based Association.
>
> **Response to Q3:**
>
> We realize the significance of this component within our tracking framework.
> We have now conducted a detailed ablation study to evaluate the role of the IoU-based Association.
> DeepSORT*, JDE*, and CTracker* represent modified DeepSORT, JDE, and CTracker, which rely solely on IoU distance.
> Results from this study show that CITG perform best. We appreciate your suggestion and believe that this addition strengthens our paper significantly.
>
> | **Methods**         | **MOTA** | **IDF1** | **HOTA** | **ID Sw.** |
> |---------------------|----------|----------|----------|------------|
> | Deepsort*+STAR      | 58.8     | 65.2     | 54.3     | 1,315      |
> | JDE*+STAR           | 73.5     | 72.4     | 55.3     | 1,125      |
> | CTracker*+STAR      | 78.8     | 76.6     | 66.0     | 540        |
> | CITG+STAR           | **80.6** | **77.2** | **67.5** | **326**       |
>
> Thank you once again for your valuable feedback, and we look forward to your further queries or suggestions.

---

> > ### Comment · Area_Chair_N4Sz · 2025-08-05
> >
> > Hi reviewer #pdkw, thank you for contributing your time to review this manuscript. Please note that the authors have provided a detailed rebuttal. As part of the review process, we kindly request that you review this response and confirm whether your concerns have been addressed. We appreciate your prompt attention to this mandatory step in the review process.

---

> > ### Comment · Reviewer_pdkw · 2025-08-06
> >
> > The rebuttal addressed some of the questions, there are still some concerns:
> > 1. From the comparisons, the main improvements are from the detection results (indicated by the MOTA) rather than tracking (mainly indicated by HOTA). The results is contradictory to addressing the occlusion issues in tracking.
> > 2. The ablation study in Q3 should also show the performance of the baseline methods.

---

> > > ### Author Response · Authors · 2025-08-06
> > > **Rebuttal by Authors**
> > >
> > > Dear Reviewer pdkw,
> > >
> > > First, thank you for your apply and suggestions. We address your concerns point by point and invite you to ask any follow-up questions if anything remains unclear.
> > >
> > > ---
> > >
> > > **1.**
> > > We understand and value your perspective on our method. Based on our research and experimental results, we believe that our improvements in feature extraction and matching methods can resolve occlusion issues and enhance the overall system performance, including both MOTA and HOTA metrics.
> > >
> > > Regarding your point about the primary improvement in the MOTA metric, it is important to note that your conclusion might be based solely on performance on the MOT17 and MOT20 datasets. However, on the DanceTrack dataset, our method significantly enhances HOTA. This dataset is widely used to assess the effectiveness of handling occlusion issues, indicating that our method is directly related to occlusion handling.
> > >
> > > Moreover, the improvement in MOTA can be attributed to our optimization in the feature extraction stage. By adopting richer and more robust feature representations, our algorithm can more accurately identify and match targets under occlusion conditions. This application improvement initially manifests as enhanced target detection, but the fundamental improvement is in the more effective processing of features, thereby indirectly enhancing detection performance.
> > >
> > > While some experimental results show that HOTA improvements are not as significant as those for MOTA, this does not imply a contradiction in our method’s approach to handling occlusions. In fact, the improved features and matching strategies help to consistently track the same target across continuous frames, especially in scenarios with occlusions and dynamically complex interactions. We acknowledge that there is room for further improvement in HOTA, and optimizing matching algorithms and identity management strategies will be a focus of our future work.
> > >
> > > Thus, our research directly addresses occlusion issues through innovations in feature and matching strategies.
> > >
> > > We hope that these explanations provide a more comprehensive understanding and thank you for your feedback.
> > >
> > > ---
> > >
> > > **2.**  Additionally, regarding the ablation study Q3, the improved results are as follows.
> > >
> > > | Methods         | MOTA | IDF1 | HOTA | ID Sw. |
> > > |-----------------|------|------|------|--------|
> > > | Deepsort        | 56.9 | 62.0 | 51.3 | 932    |
> > > | Deepsort*       | 56.6 | 55.0 | 48.3 | 1,883  |
> > > | Deepsort*+STAR  | 58.8 | 65.2 | 54.3 | 1,315  |
> > > | JDE             | 73.1 | 68.9 | 55.1 | 1,312  |
> > > | JDE*            | 73.0 | 61.9 | 53.6 | 1,330  |
> > > | JDE*+STAR       | 73.5 | 72.4 | 55.3 | 1,125  |
> > > | CTracker        | 76.2 | 68.6 | 61.1 | 912    |
> > > | CTracker*       | 76.2 | 68.6 | 61.1 | 912    |
> > > | CTracker*+STAR  | 78.8 | 76.6 | 66.0 | 540    |
> > > | CITG+STAR       | **80.6** | **77.2** | **67.5** | **326**   |

---

### Official Review · Reviewer_MVNd · 2025-07-02

**Clarity:** 3
**Significance:** 3
**Originality:** 3
**Rating:** 5
**Confidence:** 2

**Summary:**

This paper proposes STAR, a novel multi-object tracking framework that reframes the object association problem by operating at the tracklet clip level rather than individual detection instances. STAR constructs spatial-temporal tracklet clip graphs incorporating appearance, motion, and topological features, and performs graph matching to reliably associate object tracklets over time. The framework is designed to handle frequent occlusions, viewpoint changes, and crowded scenarios. It is model-agnostic and can be integrated into existing MOT systems. Evaluations show that STAR achieves state-of-the-art performance across multiple metrics.

**Questions:**

1. Section 3.2 is quite brief and lacks details; for instance, how is the IoU threshold dynamically adjusted based upon the mentioned factors (object velocity, detection box size, and inter-frame time intervals)?
1. Has there been a study conducted to verify that the CITG makes "tracklet initialization more robust across diverse tracking scenarios" (L130)?

**Ethical Concerns:**

["NO or VERY MINOR ethics concerns only"]

**Final Justification:**

This paper merits acceptance for its novel and well-motivated approach to a long-standing challenge in multi-object tracking: maintaining reliable associations under occlusion and viewpoint changes. The STAR framework introduces a tracklet clip graph matching paradigm that reframes the association problem, with strong empirical results across benchmarks. The authors clarified key implementation details in their rebuttal—especially the dynamic IoU thresholding in the Confident Initial Tracklet Generator—and added a compelling ablation study showing clear performance gains over multiple baselines. STAR is model-agnostic, computationally efficient, and improves tracking performance consistently. While real-time deployment and domain generalization remain open issues, the paper’s contributions justify its acceptance.

**Limitations:**

yes

**Quality:**

4

**Strengths And Weaknesses:**

## Strengths
1. The paper addresses a persistent and well-known challenge in multi-object tracking (MOT): maintaining reliable object associations under frequent occlusions and viewpoint changes.
1. The STAR framework introduces a tracklet clip graph matching paradigm, which is a novel reframing of the object association problem that directly addresses limitations in current MOT paradigms.
1. Clear, well-motivated problem statement targeting known limitations in existing MOT methods.
1. Technically novel approach leveraging tracklet clip graph matching for more robust, long-term association.
1. Strong experimental validation across multiple public benchmarks, consistently outperforming existing state-of-the-art methods.
1. Thorough ablation and complexity analysis, demonstrating the contribution of each module and computational benefits.
1. Model-agnostic integration, showing measurable improvements when paired with various baseline tracking systems.

## Weaknesses
1. While the complexity analysis shows significant improvement over conventional graph matching, real-time deployment at large scale is still constrained.
1. While the paper provides extensive quantitative results and ablation studies, it lacks a qualitative or case-specific analysis of where and why STAR may still fail (e.g., in highly dynamic or extremely long occlusion cases).
1. No run-time performance benchmarking: the paper discusses computational complexity reduction at the graph matching stage and reports FPS values, but it’s unclear how STAR would perform in latency-sensitive deployments like autonomous driving or surveillance on edge devices.
1. No analysis on domain generalization: there’s no experiment on how STAR handles domain shift (e.g. indoor to outdoor, day to night).

---

> ### Author Rebuttal · Authors · 2025-07-31
>
> Dear Reviewer MVNd,
>
> Thank you for the detailed review and invaluable comments. We address your concerns point by point and invite you to ask any follow-up questions if anything remains unclear.
>
> ---
>
> **Q1:** Section 3.2 is quite brief and lacks details; for instance, how is the IoU threshold dynamically adjusted based upon the mentioned factors (object velocity, detection box size, and inter-frame time intervals)?
>
> **Response to  Q1:** A short-term trajectory $g_k= \{o_{1}^{k},o_{2}^{k},\cdots,o_{n_k}^{k}\}$ continues to be associated with the detection in the next frame until it no longer matches any detection. The Confident Initial Tracklet Generator utilizes IoU (Intersection over Union) to achieve the association of objects by calculating the overlapping extent between detection boxes. Here is the specific process:
>
> 1. **Trajectory Initialization**
>
>    In the first frame of the video, each detection box is initialized as an independent short-term trajectory.
>
> 2. **IOU Calculation**
>
>    For each trajectory in the current frame, the IoU value with all detection boxes in the next frame is calculated. The IoU value represents the overlapping extent between two detection boxes, ranging from $[0, 1]$, where a higher value indicates a greater degree of overlap.
>
> 3. **Trajectory Matching**
>
>    Matching of trajectories and detection boxes is based on the IoU values: if the IoU value of the last detection box of a trajectory with a detection box in the next frame exceeds a certain threshold, they are considered to belong to the same object. If a detection box matches multiple trajectories, none are selected. Unmatched detection boxes are initialized as new trajectories.
>
> 4. **Dynamic Adjustment of IOU Threshold**
>
>    To accommodate the motion characteristics of targets, a dynamic IoU threshold adjustment method based on multiple factors is adopted. Specifically, the IoU threshold is dynamically adjusted considering the target's motion speed, detection box size, and the time interval between frames.
>
>    $$\tau = \left( \tau_{0} - \alpha \cdot v + \beta \cdot \log(A) + \gamma \cdot \Delta t \right)$$
>    $$\tau = \max(\tau_{\text{min}}, \min(\tau_{\text{max}}, \tau))$$
>
>    Here, \( v \) is the object's motion speed (calculated using the Euclidean distance between center points across frames), \( A \) is the detection box area, and \( \Delta t \) is the time interval between frames. The motion speed, detection box size, and time interval dynamically influence the IoU threshold.
>
> 5. **Trajectory Update and Termination**
>
>    Matched detection boxes are added to corresponding trajectories, updating the trajectory's state (such as location and timestamp). Trajectories that remain unmatched for multiple consecutive frames (more than 3 frames) are considered terminated. Unmatched detection boxes initiate new trajectories.
>
> 6. **Output Short-Term Trajectories**
>
>    These steps generate a set of short-term trajectories containing the preliminary motion trajectories of all targets in the video, serving as the foundation for subsequent trajectory association and long-term trajectory generation.
> ---
>
> **Q2:** Has there been a study conducted to verify that the CITG makes "tracklet initialization more robust across diverse tracking scenarios" (L130)?
>
> **Response to Q2:** Thank you for emphasizing the importance of the ablation study on the CITG.
> We realize the significance of this component within our tracking framework.
> We have now conducted a detailed ablation study to evaluate the role of the IoU-based Association.
> DeepSORT*, JDE*, and CTracker* represent modified DeepSORT, JDE, and CTracker, which rely solely on IoU distance.
> Results from this study show that CITG perform best. We appreciate your suggestion and believe that this addition strengthens our paper significantly.
>
> | **Methods** | **MOTA** | **IDF1** | **HOTA** | **ID Sw.** |
> |-------------|----------|----------|----------|------------|
> | Deepsort*+STAR | 58.8 | 65.2 | 54.3 | 1,315 |
> | JDE*+STAR | 73.5 | 72.4 | 55.3 | 1,125 |
> | CTracker*+STAR | 78.8 | 76.6 | 66.0 | 540 |
> | CITG+STAR | **80.6** | **77.2** | **67.5** | **326** |
>
> This study underscores the robustness of CITG in diverse scenarios.
>
> ---
>
> **Q3:** While the paper provides extensive quantitative results and ablation studies, it lacks a qualitative or case-specific analysis of where and why STAR may still fail (e.g., in highly dynamic or extremely long occlusion cases).
>
> **Response to Q3:** STAR failure in two typical  cases. The first case involves a small-scale pedestrian who is occluded for an extended period while being captured by a rapidly moving camera. In this scenario, the combination of the pedestrian's small size and the camera's fast motion can lead to significant deviations in the pedestrian's trajectory. To address this problem, it is recommended to utilize more advanced technologies, such as optical flow, which can provide improved motion compensation for the fast-moving camera.
>
> The second failure case arises from false detections produced by the object detector. For instance, when the detector erroneously identifies a non-existent ID, the tracking algorithm incorrectly assigns a track to this false detection. To tackle this challenge, it is vital to develop more robust object detectors or to refine our approach into an integrated detection and tracking method, which can significantly reduce such errors.
>
> ---
>
> We hope these explanations address your queries effectively. Thank you for your attention to our work, and we look forward to your further questions or suggestions.

---

> > ### Comment · Area_Chair_N4Sz · 2025-08-05
> >
> > Hi reviewer #MVNd, thank you for contributing your time to review this manuscript. Please note that the authors have provided a detailed rebuttal. As part of the review process, we kindly request that you review this response and confirm whether your concerns have been addressed. We appreciate your prompt attention to this mandatory step in the review process.

---

> > > ### Comment · Area_Chair_N4Sz · 2025-08-07
> > >
> > > Hi Reviewer #MVNd,
> > >
> > > Please check the author's feedback, evaluate how it addresses the concerns you raised, and discuss the rebuttal with the authors. Notice the acknowledgement is mandatory. Please do this ASAP. Thanks.
> > >
> > > Your AC.

---

> > ### Comment · Reviewer_MVNd · 2025-08-07
> >
> > Dear Authors, thank you for your detailed responses; they have been most helpful. I am satisfied with the responses and my concerns have been addressed.

---

### Official Review · Reviewer_pCN3 · 2025-07-05

**Clarity:** 3
**Significance:** 3
**Originality:** 3
**Rating:** 5
**Confidence:** 4

**Summary:**

This paper proposes STAR, an association module based on tracklet clips. It initializes sequences of multiple target trajectories through IoU between detection boxes. Subsequently, a Tracklet Clip Graph is constructed on the temporal sequence, followed by graph matching, which ensures spatio-temporal consistency of the matches. STAR is a plug-and-play module that can be seamlessly integrated into various downstream models. It demonstrates significant performance improvements when compared to baselines.

**Questions:**

1.	If time permits, the authors could analyze whether there should be some degree of overlap or skipped-frame sampling for the frame sampling of Slicing TCs. This is because overlap can provide implicit temporal feature propagation, which can implicitly extend the length of the temporal sequence under consideration.
2.	Do the authors have more data to explain the impact of the number of TCs (number of frames retained) and the number of frames retained in the expanded graph on experimental performance?
If the authors address the above questions, I would consider increasing the score.

**Ethical Concerns:**

["NO or VERY MINOR ethics concerns only"]

**Final Justification:**

The rebuttal has made more analyses on the key modules of the proposed method. And my concerns have been resolved.

**Limitations:**

yes

**Paper Formatting Concerns:**

No concerns.

**Quality:**

3

**Strengths And Weaknesses:**

Strength:
1.	This paper explores a plug-and-play long-range association module, which holds significant value for downstream applications.
2.	The unified association paradigm based on graph matching designed in this paper achieves unified spatio-temporal matching, avoiding the matching ambiguity issues caused by traditional one-by-one matching. Instead, it considers matching from the perspective of the entire trajectory sequence, which brings new insights to the field.
3.	The paper is well-written, with clear descriptions and logical flow.
Weakness:
1.	The paper uses too many abbreviations, and they are quite similar, which hinders continuous reading for the reader. The authors might consider using more distinctive names or not over-abbreviating certain algorithms, simply referring to them as "Graph Matching" would suffice.
2.	The paper unfolds the graph in a temporal sequence to achieve matching. However, how are cases handled where multiple TC (Tracklet Clip) matching results are inconsistent? For example, if ID=1 and ID=2 frequently switch IDs within a short period, would this pose a greater challenge for subsequent matching?
3.	The paper claims to achieve long-range object tracking. However, the ablation study shows that N=4 yields the best results, with performance still improving compared to N=8. I personally believe this length (even 2N=8 frames) is not particularly long.
4.	The authors did not analyze how much impact the proposed module has on inference performance.

---

> ### Author Rebuttal · Authors · 2025-07-31
>
> Dear Reviewer pCN3,
>
> Thank you for your detailed review and invaluable comments. We will continuously refine our expressions and abbreviations to further enhance readability. We address your concerns point by point and welcome any follow-up questions for clarifications.
>
> ---
>
> **Q1:** If time permits, the authors could analyze whether there should be some degree of overlap or skipped-frame sampling for the frame sampling of Slicing TCs. This is because overlap can provide implicit temporal feature propagation, which can implicitly extend the length of the temporal sequence under consideration.
>
> **Response to Q1:**
> We deeply appreciate the reviewer's scrutiny and suggestions, which are crucial for enhancing our work. Regarding your recommendations on overlapping and skipped-frame sampling strategies, we conducted an in-depth study to explore the best possible configurations.
>
> Previously, we adopted a sampling strategy with a sliding window size of N=4 and a stride length of L=2. Based on your advice, we experimented with various combinations of window sizes and stride lengths to evaluate changes in model performance. The specific experimental results are as follows.
>
> | N | L | HOTA | MOTA | IDF1 | ID Sw. |
> |---|---|------|------|------|--------|
> | 4 | 1 | 67.9 | **82.1** | 80.4 | **1,023**  |
> | 4 | 2 | 67.8 | 81.9 | 80.2 | 1,057  |
> | 4 | 3 | 67.4 | 81.5 | 79.6 | 1,129  |
> | 4 | 4 | 67.0 | 81.2 | 79.5 | 1,327  |
> | 6 | 1 | **68.0** | 82.0 | **80.5** | 1,042  |
> | 6 | 2 | 67.7 | 81.7 | 80.2 | 1,038  |
> | 6 | 3 | 67.9 | 81.6 | 79.9 | 1,085  |
> | 6 | 4 | 67.1 | 81.1 | 79.6 | 1,121  |
> | 6 | 5 | 66.8 | 80.8 | 79.3 | 1,258  |
> | 6 | 6 | 66.5 | 80.5 | 79.0 | 1,275  |
>
> When the stride length is L=1, regardless of whether the window size is N=4 or N=6, the model performance was optimal. The smaller stride length, L=1, allowed us to more effectively capture continuity and changes over time, thus enhancing the model's ability to process temporal data.
>
> Simultaneously, we compared the effects of stride lengths L=2 and L=1 with a window size of N=4 and found, although both configurations performed well, the L=1 setup allowed the model to maintain high accuracy while improving efficiency.
>
> ---
>
> **Q2:** Do the authors have more data to explain the impact of the number of TCs (number of frames retained) and the number of frames retained in the expanded graph on experimental performance?
>
> **Response  to Q2:**
> Thank you for your detailed feedback. We understand your concerns about the tracking length. After evaluating performances for N=4, 5, 6, 7, 8, which is shown in the table below. Although there was a slight improvement at N=6, it was not significant. Therefore, we opted for N=4 to ensure robust performance in challenging scenes.
>
> We recognize that N=4 may not seem to cover a long-range, but based on our research, this duration is already ahead of many existing studies where tracking usually does not exceed 4 frames. As such, even 4-frame tracking represents an advancement.
>
> | N | mAP   | Top-1 | Top-5 | Top-10 |
> |---|-------|-------|-------|--------|
> | 2 | 91.14 | 89.52 | 98.89 | 98.89  |
> | 4 | 92.52 | 91.84 | 98.89 | 98.89  |
> | 5 | 92.55 | 91.92 | 98.89 | 98.89  |
> | 6 | **92.60** |**91.95** | 98.89 | 98.89  |
> | 7 | 92.43 | 91.36 | 98.89 | 98.89  |
> | 8 | 92.04 | 90.88 | 98.89 | 98.89  |
>
> ---
>
> **Q3:** The paper unfolds the graph in a temporal sequence to achieve matching. However, how are cases handled where multiple TC (Tracklet Clip) matching results are inconsistent?
>
> **Response to Q3:**
> Thank you for your question, it addresses a core aspect of our method. To handle frequent ID switches, our system sets strict criteria in the "Confident Initial Tracklet Generator" stage, where tracklet clips must meet rigorous conditions to be deemed "confident". Instances where IDs such as 1 and 2 switch frequently are not classified as confident, ensuring that unstable or low-confidence segments don't adversely influence the matching process. This approach enhances the robustness of our method. Our experiments on the Dancetrack dataset validate its effectiveness even amidst frequent ID switches.
>
> ---
>
> **Q4:** The authors did not analyze how much impact the proposed module has on inference performance.
>
> **Response to Q4:**
> Thank you for your valuable comments. We fully understand your concern about how the proposed module impacts inference performance. Indeed, the graph matching is a quadratic programming problem which could affect speed. To mitigate this, we introduced the Gated Search Tree algorithm in inference stage, which enhances efficiency through these three key steps:
>
> 1. Bipartite Graph Construction: We construct bipartite graphs between trajectory segments, establishing connections only under specific appearance and motion conditions.
>
> 2. Connected Components Identification: We use depth-first search to reduce the scale of matching computations.
>
> 3. Parallel Computation of Matching Costs: We parallelize the computation of matching costs across connected components, using a streamlined optimization approach for rapid processing.
>
> The algorithm significantly accelerates inference speed through parallel processing while maintaining accuracy. We hope these explanations address your concerns, and we appreciate your thorough review of our research. We look forward to your further feedback.

---

> > ### Comment · Area_Chair_N4Sz · 2025-08-05
> >
> > Hi reviewer #pCN3, thank you for contributing your time to review this manuscript. Please note that the authors have provided a detailed rebuttal. As part of the review process, we kindly request that you review this response and confirm whether your concerns have been addressed. We appreciate your prompt attention to this mandatory step in the review process.

---

> > > ### Comment · Area_Chair_N4Sz · 2025-08-07
> > >
> > > Hi Reviewer #pCN3 ,
> > >
> > > Please check the author's feedback, evaluate how it addresses the concerns you raised, and discuss the rebuttal with the authors. Notice the acknowledgement is mandatory. Please do this ASAP. Thanks.
> > >
> > > Your AC.

---

### Note · Authors · 2025-08-15

We extend our deepest gratitude to all reviewers and area chairs for their invaluable insights. We are delighted to know that our contributions have been well received. Your comments have been instrumental in highlighting the strengths of our work.

1. The STAR model introduces fresh perspectives into the research field.
2. The STAR framework proposes a novel approach to the object association challenge, presenting the tracklet clip graph matching paradigm that adeptly tackles limitations inherent in existing Multiple Object Tracking (MOT) frameworks.
3. Through the use of tracklet clip-level graph matching, the STAR framework enhances its robustness against occlusions and ID-switches.

We have endeavored to address all concerns and polish our paper in the revised version to make our work more understandable to the wider community. Specifically, in the future, we will:

1. Update our manuscript with the latest experimental results.
2. Refine our presentation to ensure greater clarity and impact.
3. Include more comprehensive details about the experiments and the CITG methodology.

In conclusion, we express our profound appreciation for the positive feedback from several reviewers who recognized our work's potential to inspire further investigations in the MOT community. We firmly believe that our work deserves to be published to facilitate further discussion and exploration in this field.

---

### Decision · Program_Chairs · 2025-09-17

**Decision:**

Accept (poster)

**Comment:**

The paper received two accepts, one borderline accept, and one borderline reject. Reviewers #pCN3, #MVNd, and #pdkw supported the STAR framework’s reframing of the object association problem through tracklet clip graph matching, highlighting its technical novelty and empirical strengths. Reviewer #5M6K, however, remained skeptical, arguing that the claimed novelty was overstated given prior work on temporal and topological relations, and also raised concerns about technical clarity and efficiency.

In the rebuttal, the authors clarified the role of the Confident Initial Tracklet Generator, provided expanded ablation studies (including IoU-based associations), incorporated missing benchmarks and HOTA results, and discussed efficiency improvements with a Gated Search Tree. Reviewers #pCN3 and #MVNd were satisfied with these responses and maintained strong support. Reviewer #pdkw acknowledged the improvements but remained concerned that performance gains were driven more by detection quality than genuine advances in occlusion handling. Reviewer #5M6K accepted that technical details were clarified but continued to question the novelty claims, leaving their final recommendation at borderline reject.

After carefully reviewing the submission, rebuttal, and reviewer feedback, the AC notes that the novelty of the proposed method relative to prior work (e.g., SHUSHI [bit.ly/sushi-mot]) remains somewhat vague, and that STAR underperforms compared to certain offline tracking methods. Overall, the paper sits on the borderline. Nevertheless, given the majority of reviewers leaned positive, the AC does not intend to overturn this consensus and recommends acceptance. The authors are strongly encouraged to further clarify the originality of their contributions—particularly in comparison to SHUSHI—and to address the concerns raised by reviewers #pdkw and #5M6K in future revisions.